# Preoperative Lower-Limb Muscle Predictors for Gait Speed Improvement after Total Hip Arthroplasty for Patients with Osteoarthritis

**DOI:** 10.3390/jpm13081279

**Published:** 2023-08-20

**Authors:** Tadashi Yasuda, Satoshi Ota, Sadaki Mitsuzawa, Shinnosuke Yamashita, Yoshihiro Tsukamoto, Hisataka Takeuchi, Eijiro Onishi

**Affiliations:** Department of Orthopaedic Surgery, Kobe City Medical Center General Hospital, 2-1-1 Minatojimaminami-machi, Chuo-ku, Kobe 650-0047, Japan

**Keywords:** computed tomography, gait speed, hip abductor, hip joint, muscle composition, total hip arthroplasty

## Abstract

This study aimed to identify preoperative lower-limb muscle predictors for gait speed improvement after total hip arthroplasty (THA) with hip osteoarthritis. Gait speed improvement was evaluated as the subtraction of preoperative speed from postoperative speed. The preoperative muscle composition of ipsilateral hip abductors was evaluated using computed tomography. The females (*n* = 45) showed smaller total cross-sectional areas of the gluteal muscles than the males (*n* = 13). The gluteus maximus in the females showed lower lean muscle mass area (LMM) and higher ratios of the intramuscular fat area and the intramuscular adipose tissue area to the total muscle area (TM) than the males. Regression analysis revealed that LMM/TM of the glutei medius and minimus may correlate negatively with postoperative improvement in gait speed. Receiver operating characteristic curve analysis for prediction of minimum clinically important improvement in gait speed at ≥0.32 m/s resulted in the highest area under the curve for TM in the upper portion of the gluteus maximus with negative correlation. The explanatory variables of hip abductor muscle composition predicted gait speed improvement after THA more precisely in the females compared with the total group of both sexes. Preoperative muscle composition should be evaluated separately based on sex for the achievement of clinically important improvement in gait speed after THA.

## 1. Introduction

Osteoarthritis (OA) is a major cause of disability. The goal of total hip arthroplasty (THA) for patients with hip OA is to improve physical function, which is commonly observed within the first 6 months [1]. However, approximately 10% of patients after THA report insufficient functional recovery [2]. Because postoperative walking ability is a critical factor for independent daily activities, gait function after THA is highly associated with patients’ satisfaction with postoperative outcomes. As the core set of performance measures for OA patients, the assessment of gait speed is recommended by the Osteoarthritis Research Society International guidelines [3]. A recent study to identify clinically meaningful benchmarks for gait improvement after THA has shown that the minimum clinically important improvement (MCII) in gait speed after THA is 0.32 m/s [4]. Currently, it remains to be investigated which preoperative factors can be linked to the suboptimal recovery of gait function after THA.

Hip abductor muscles stabilize the pelvis, maintain the level of the contralateral pelvis, and prevent hip adduction during single-leg stance as the basis of human locomotion [5]. The abductor muscle function is directly associated with physical function after THA [6]. Hip abductor muscles can be divided into superficial muscles that offer their effect via insertion into the iliotibial band and deeper muscles that work via insertion into the greater trochanter. While the superficial muscles contain the upper portion of the gluteus maximus and the tensor fascia lata, the deep muscles include the gluteus medius, the gluteus minimus, and the piriformis [7]. Each muscle of hip abductors can be evaluated by computed tomography (CT). In addition to the cross-sectional area measurement, CT has been employed for accurate estimates of structural muscle composition. High-density lean tissues or lean muscle mass (LMM), low-density lean tissues, and intra-muscular fat are evaluated as components of muscle composition [8]. There is a high association between muscle strength and LMM [9]. Age-related loss of LMM results in a loss of muscle strength [10]. In elderly people, the adipose tissue beneath the deep fascia of a muscle, intramuscular adipose tissue (IMAT), may cause muscle strength deficit because an altered orientation of muscle fibers by fatty infiltration into skeletal muscle can reduce the force-producing capacities [11]. 

There are age-associated differences in body composition such as skeletal muscle mass and fat distribution between sexes [12]. A series of studies have demonstrated that females have a higher risk and prevalence rate of hip OA than males [13,14,15]. However, no information is available about potential sex effects on hip abductor muscle composition before THA or on gait function after THA. Therefore, this study aimed to clarify hip abductor muscle composition in female patients with hip OA. There could be sex-related differences in the preoperative composition of ipsilateral hip abductor muscles. This study was also conducted to identify preoperative lower-limb muscle predictors in association with the MCII in gait speed after THA. In female patients, the cross-sectional area of the upper portion of the gluteus maximus may predict the MCII in gait speed after THA.

## 2. Materials and Methods

### 2.1. Patient Selection

We retrospectively analyzed the data from 126 OA patients who underwent primary THA between 2019 and 2020 in our hospital. The exclusion criteria were a history of total knee arthroplasty (11 patients), contralateral THA within 6 months before admission (11 patients), a hip surgical procedure on the operated side (9 patients), hip deformity with Crowe types 2, 3, and 4 (19 patients) [16], bilateral THA (3 patients), contralateral pain with hip or knee OA (10 patients), and insufficient clinical data (5 patients). As a result, 58 of 126 patients (45 females and 13 males) were enrolled in this study (Figure 1). Of the 58 patients, there were 10 patients who had undergone contralateral THA more than 12 months before admission. All the patients were able to walk independently with or without a cane before surgery.

### 2.2. Operation and Postoperative Rehabilitation

We employed a lateral approach with a modified Mostardi technique for minimal damage to the hip abductor muscles [17] to perform THA. Following blunt dissection through the anterior one-fourth of the gluteus medius, the bony portion that retains the tendinous junction of the gluteui medius and minimus was osteotomized using a chisel. The osteotomized trochanteric fragment, approximately 10 mm long × 10 mm wide × 5 mm deep, was mobilized anteriorly and medially, whereas the vastus lateralis was kept completely intact. Finally, the osteotomized trochanteric fragment was anatomically reattached by inducing bone-to-bone contact using ultrahigh molecular-weight polyethylene sutures. The same rehabilitation protocol was offered for each inpatient during the first 2 weeks in our hospital and thereafter 3–4 weeks in the recovery-phase rehabilitation hospital after operation. A standard rehabilitation program was initiated on the first postoperative day, and patients were allowed to eliminate walking aids whenever comfortable. Physical therapy included progressively improving walking ability, other functional activities, and walking stairs according to the needs and progress of each patient. Patients took part in a progressive program involving a range of motion exercises, strengthening exercises, and functional training. No patient received outpatient physical therapy.

### 2.3. Muscle Composition

Muscle composition of the operated limb was estimated by CT taken for preoperative planning within three weeks before THA. Muscle composition of the glutei medius and minimus and the gluteus maximus was evaluated on a single axial CT slice at the bottom end of the sacroiliac joint [18,19]. The inferior point of the sacroiliac joint is found to be the appropriate site for the measurement of the cross-sectional area of the gluteus medius in patients with hip OA because the area at this level significantly correlates with muscle volume and muscle strength [19]. In addition, the upper portion of the gluteus maximus was analyzed almost exclusively at this level because the upper portion arises from the posterior iliac crest whereas the lower portion originates from the inferior sacrum and upper lateral coccyx [7]. The position of the pelvis was standardized during imaging by ensuring that the line connecting the anterior superior iliac crest on both sides was perpendicular to the bed [19].

The muscle groups were manually outlined and thereafter automatically segmented based on attenuation values: −29 to 150 Hounsfield units (HU) using SYNAPSE VINCENT software version 5.0 (Fujifilm Co., Tokyo, Japan), according to the previous study [18]. The software electronically calculated the cross-sectional area in cm^2^ of the segmented total muscle group (TM). High-density lean tissue that comprises little fatty infiltration is recognized as a measure of LMM. Elevated levels of adipocytes between and within muscle fibers in low-density lean tissue (LDL) result in decreased CT density compared with LMM. Intramuscular fat (mFAT) shows the lowest CT density. Cross-sectional areas in cm^2^ of LMM, LDL, and mFAT within each TM were measured electronically with the software as the areas of pixels according to the definition by attenuation values: 30 to 80 HU for LMM, 0 to 29 HU for LDL, and −190 to −30 HU for mFAT [8,18]. Representative CT images for muscle composition measurement are shown in Figure 2. IMAT was defined as the summation of the areas of both LDL and mFAT [8,18]. As shown in the previous study [18], the intra-class correlation coefficient (ICC) was 0.98–0.99 for muscle composition measurement. The area of TM or each component was normalized for the square of the patient’s height (cm^2^/m^2^). Alternatively, LMM, LDL, mFAT, and IMAT were normalized for the respective muscle’s size by calculating a percentage of each measure relative to TM [8,18], designated as LMM/TM, LDL/TM, mFAT/TM, and IMAT/TM. Because alterations in the place of muscle section by hip deformity may affect cross-sectional CT analysis, this study excluded patients with hip deformity of Crowe types 2, 3, and 4.

### 2.4. Functional Outcome Measures

The following measures were assessed at admission and at 6 months postoperatively when the functional performance reached a plateau after THA [1]. Self-selected comfortable gait speed was evaluated by timing a patient instructed to walk at his or her routine speed across a 10 m course with 2 m acceleration and deceleration zones [20]. Gait speed (m/s) was measured twice, and the faster speed was used for analysis. Postoperative improvement in gait speed was calculated as the subtraction of preoperative speed from postoperative speed. Isometric muscle strength of the hip abductor and knee extensor was measured bilaterally using a handheld dynamometer as described previously [21]. ICC ranged from 0.93 to 0.96 for the measurement of maximal isometric strength.

### 2.5. Statistical Analysis

The normality of data was evaluated by Shapiro–Wilk test. Comparisons were performed between the female and male data by *t* test and Mann–Whitney U test for parametric and nonparametric data, respectively. Effect size (r) was calculated for the association between the variables. Pearson’s correlation coefficients were also calculated to determine associations of preoperative variables with postoperative improvement in gait speed. Stepwise multiple regression analysis was used to identify significant explanatory variables for postoperative improvement in gait speed. Logistic regression analyses and receiver operating characteristic (ROC) curve analyses were performed using the cutoff of the MCII of gait speed at 0.32 m/s [4]. Differences were considered to be statistically significant at *p* < 0.05. All analyses were performed using the SPSS software package (version 28.0; IBM SPSS Statistics, Chicago, IL, USA). A power analysis showed that at least 40 patients were required to perform multiple regression analysis with an effect size of 0.35, a power of 0.80, and an alpha error of 0.05 using 4 predictors.

## 3. Results

### 3.1. Demographics, Gait Speed, and Muscle Strength

Data of the 45 female patients, the 13 males, and the total of both sexes are shown separately in Table 1.

There was no difference in age between the female and the male patients. Body mass index (BMI) was higher in the male group compared with the female one. No difference between the sexes was found in gait speed before and after THA as well as in the postoperative speed improvement. While there was no difference in the knee extensor strength of the operated limb before THA between the sexes, the postoperative strength was stronger in the male patients than in the females. The preoperative and postoperative knee extensor strength of the non-operated limb was higher in the male group compared with the female one. No difference was found between the sexes in the ipsilateral and contralateral hip abductor strength before and after THA.

### 3.2. Preoperative Muscle Composition of the Operated Limb

The preoperative muscle composition of the glutei medius and minimus and the upper portion of the gluteus maximus in the 45 female patients, the 13 males, and the total of both sexes are shown individually in Table 2.

TM of the glutei medius and minimus was larger in the male patients compared with the females. There was no difference in other compositions of the glutei medius and minimus between the sexes. TM and LMM of the gluteus maximus were larger in the male group than the female one. mFAT/TM and IMAT/TM of the gluteus maximus were higher in the female patients compared with the males. No difference was found in other components of the gluteus maximus between the sexes.

### 3.3. Correlations of Gait Speed Improvement with Muscle Composition and Strength

Pearson’s correlation coefficients of gait speed improvement with preoperative muscle composition and strength are demonstrated separately in the female patients and in the total of both female and male patients in Table 3. Pearson’s correlation coefficients of gait speed at admission and at 6 months after THA are also shown in Table 3 for reference. In the total group, LMM/TM and LDL of the glutei medius and minimus were associated negatively and positively with postoperative improvement in gait speed, respectively. In the female group, LMM and LMM/TM of the glutei medius and minimus correlated positively with postoperative improvement in gait speed. LDL/TM of the glutei medius and minimus showed a positive association with postoperative speed improvement. In addition, there was a negative correlation between TM of the gluteus maximus and postoperative speed improvement. No association was found between preoperative muscle strength and postoperative improvement in gait speed in the total or the female group.

Next, stepwise regression analyses were conducted to identify independent variable(s) for postoperative improvement in gait speed. Table 4 demonstrates the results of stepwise regression analyses in the female patients and in the total of both female and male patients individually. From regression analysis in the total group using the two significant variables of muscle composition for postoperative improvement in gait speed (Table 3), LMM/TM of the glutei medius and minimus was selected as an independent variable. When regression analysis was performed in the female group using the four significant variables of muscle composition for gait speed improvement after THA (Table 3), LMM/TM of the glutei medius and minimus was also selected as an independent variable.

### 3.4. Preoperative Predictors for the MCII in Gait Speed

This study employed the cutoff of 0.32 m/s for the MCII in gait speed after THA [4]. For the total number of patients of both sexes, there were 39 and 19 patients with gait speed improvement at < 0.32 m/s and at ≥ 0.32 m/s at 6 months after THA, respectively. Logistic regression analyses using preoperative muscle composition selected TM and LMM of the gluteus maximus as preoperative predictors for the MCII in gait speed (Table 5). ROC curve analyses showed TM of the gluteus maximus with the highest area under the curve (AUC) of 0.700 (Table 5). With regard to the cut-off value based on the ROC curve, TM of the gluteus maximus demonstrated 87.2% specificity and 57.9% sensitivity with a cutoff at 8.1 cm^2^/m^2^ (Figure 3). In the female patients, there were 32 and 13 patients with postoperative improvement in gait speed at <0.32 m/s and at ≥0.32 m/s, respectively. Logistic regression analysis using preoperative muscle composition revealed that the TM and LMM of the gluteus maximus were preoperative predictors for the MCII in gait speed (Table 5). In addition, the LMM, LMM/TM, LDL/TM, mFAT, mFAT/TM, IMAT, and IMAT/TM of the glutei medius and minimus were found to be preoperative predictors for the MCII in gait speed (Table 5). ROC curve analyses demonstrated the TM of the gluteus maximus with the largest AUC of 0.813 (Table 5). The TM of the gluteus maximus showed 87.5% specificity and 76.9% sensitivity with a cutoff at 8.1 cm^2^/m^2^ (Figure 3). In contrast to preoperative muscle composition, preoperative muscle strength showed no predictive power for the MCII in gait speed at 6 months after THA (Table 5).

## 4. Discussion

There is increasing evidence of muscle fiber type differences and susceptibility for disuse atrophy between males and females [22]. Compared with males, females likely depend on oxidative metabolism [23] and have a greater content of type I muscle fibers within the same muscle [24,25]. Oxidative muscle fibers are preferentially affected by disuse atrophy [26,27]. Patients with hip OA demonstrate decreased volume of the gluteus maximus, the gluteus medius, and the gluteus minimus on the affected side compared with matched controls [28]. Decreased muscle volume can be a result of functional disuse of those muscles [28]. In addition to atrophy, muscle disuse can cause fatty infiltration [29]. Actually, increased fatty infiltration is found in the gluteus maximus and the gluteus minimus in the hip joint with OA [28]. This study has expanded the previous findings and is the first to suggest potential differences in preoperative muscle composition of ipsilateral hip abductors between female and male patients with hip OA. The female group showed a smaller TM of the gluteal muscles than the male one. The gluteus maximus in the female patients contained decreased LMM and increased mFAT/TM and IMAT/TM compared with that in the males. In terms of fat distribution between sexes, females likely accumulate subcutaneous fat in the lower extremities, while males tend to deposit visceral fat in the abdominal region [30]. Like subcutaneous fat, there is a possibility that fatty infiltration may increase into the upper portion of the gluteus maximus in female patients with hip OA. In contrast to the gluteus maximus, no difference was found between the sexes in the fat infiltrate in the glutei medius and minimus. Sex-related differences in fatty infiltration into different muscles remain to be clarified.

Since gait is one of the most basic functions after THA [31,32], gait speed has recently received attention as a critical factor to predict functional prognosis after THA [4,33]. Greater postoperative improvement in gait speed is associated with better clinical outcomes or greater improvement in clinical outcomes [34,35,36]. The minimum clinically important improvement (MCII) has been defined as the smallest change in measurement that indicates an important improvement in a patient’s symptoms [37]. The MCII is one of the strategies to complement conventional statistical comparisons. This study has provided the first evidence that preoperative evaluation of ipsilateral hip abductor muscle composition by CT may predict the MCII in gait speed at 6 months after THA. Foucher has applied the MCII concept for the identification of clinically meaningful benchmarks for gait improvement following THA and has found that the MCII in gait speed is 0.32 m/s with a good clinical outcome [4]. His study has also shown an association between gait speed improvement after THA and BMI [4]. In this study, however, there was no correlation between gait speed improvement after THA and BMI in the female (r = −0.246) or the total (r = −0.161) group. Alternatively, postoperative gait speed improvement significantly correlated with the preoperative muscle composition of the ipsilateral hip abductor evaluated by CT. CT can offer high-quality image reconstruction and stable attenuation values that aid in image segmentation for muscle composition assessment. Similar to CT, magnetic resonance imaging (MRI) is an excellent tool to depict detailed muscle structures. Atrophy and fatty infiltration can be observed by MRI in the gluteus medius and gluteus minimus in end-stage hip OA [38]. However, a semiquantitative grading system originally described by Goutallier et al. [39] has been used for the evaluation of fatty infiltration into muscle by MRI. In contrast to MRI, CT provides quantitative measurement of muscle composition based on the HU. Thus, quantitative analysis by CT density could be suitable for the assessment of muscle composition compared with categorical grading evaluation by MRI. Because CT is commonly employed for preoperative planning of THA, the assessment of ipsilateral hip abductor composition by CT is likely to be helpful for the selection of OA patients who may and may not achieve the gait speed benchmark after THA. A current concept review indicates sex-related differences in the outcome of THA [40]. The review also highlights the importance of the assessment of sex-related factors in patients undergoing THA to improve postoperative outcomes and patient satisfaction rates and to reduce postoperative complication rates [40]. In line with this, the present results suggest that hip OA patients should be separately evaluated before THA depending on their sexes because the explanatory variables of hip abductor muscle composition predicted gait speed improvement after THA more precisely in the female group compared with the total group of both sexes. Further studies are still necessary to identify explanatory variables of preoperative hip abductor muscle composition for postoperative outcomes in male OA patients.

From the results of stepwise regression analyses, LMM/TM of the glutei medius and minimus could be associated negatively with postoperative improvement in gait speed. The gluteui medius and minimus play a critical role in gait control. The gluteus medius works to stabilize the hip and pelvic rotation during gait [41]. The gluteus minimus is responsible for the stabilization of the femoral head within the acetabulum during the gait cycle [42]. Ipsilateral glutei medius and minimus in patients with hip OA show atrophy compared with healthy individuals [43]. In addition, the glutei medius and minimus are smaller on the ipsilateral side compared with the contralateral one [44]. Replacement of normal viable muscle tissue by fatty infiltration is observed in the glutei medius and minimus in end-stage hip OA [38,45]. As already shown in elderly people [11], increased fatty infiltration with loss of LMM in the glutei medius and minimus potentially reduces the force-generating function of the whole muscle. Actually, retainment of LMM in TM of the glutei medius and minimus demonstrated a high positive association with preoperative gait speed. In the previous study by Foucher, a slower gait speed before THA was likely to attain the MCII at ≥0.32 m/s [4]. Collectively, LMM/TM of the glutei medius and minimus may offer a negative influence on the MCII in gait speed after THA.

Based on the present results, the TM in the upper portion of the gluteus maximus was the preoperative predictor of the MCII in gait speed after THA with a negative correlation. The hip abductors consist of superficial muscles such as the upper portion of the gluteus maximus with their insertion into the iliotibial band and deeper muscles like the gluteui medius and minimus with their insertion into the greater trochanter. Hip extension is restricted during the late stance phase of gait in patients with hip OA, which may cause disuse atrophy of the lower portion of the gluteus maximus, which works as the primary hip extensor [7]. Atrophy of ipsilateral glutei medius and minimus is commonly observed in patients with hip OA [28]. In contrast, the upper portion of the gluteus maximus that acts as the hip abductor demonstrates no significant atrophy [7]. A negative correlation is found between preoperative gait speed and achievement of the MCII at ≥ 0.32 m/s after THA [4]. In the present study, the TM of the gluteus maximus showed a high positive association with preoperative gait speed. Thus, the TM in the upper portion of the gluteus maximus could be selected as the preoperative predictor for the MCII in gait speed with high specificity and sensitivity.

This study has several limitations. First, this was a monocentric retrospective study. All patients received the same surgical technique and postoperative management, which could have influenced the results. It is still unclear whether other surgical approaches can provide similar results. Second, there were no data on the muscle composition on the contralateral side or the lower portion of the gluteus maximus. Our future study needs to clarify how contralateral muscle composition affects gait speed improvement after THA. Third, this study included an insufficient number of male patients. Accordingly, the results from the comparison between the female and male patients were obtained with relatively small effect sizes. The association between muscle composition before THA and gait speed improvement after THA should be investigated for male patients with hip OA. Fourth, it remains uncertain whether patients with bilateral symptomatic OA or severe deformity may demonstrate similar results. Fifth, muscle composition was analyzed on a single axial CT slice. Although the position of the pelvis was standardized during imaging according to the established method [19], measurements in axial CT images are potentially variable and may depend on the place of the section.

## 5. Conclusions

This study indicates potential sex-related differences in preoperative muscle composition in ipsilateral hip abductors in patients with hip OA. The ratio of the lean muscle mass area to the total cross-sectional area of the glutei medius and minimus of the affected limb before THA may partly explain postoperative gait speed improvement. The total cross-sectional area of the upper portion of the gluteus maximus of the affected limb before THA could predict minimum clinically important improvement in gait speed after THA. Preoperative muscle composition should be evaluated separately depending on the sex for identification of OA patients who may gain clinically important improvement in gait speed after THA.

## Figures and Tables

**Figure 1 jpm-13-01279-f001:**
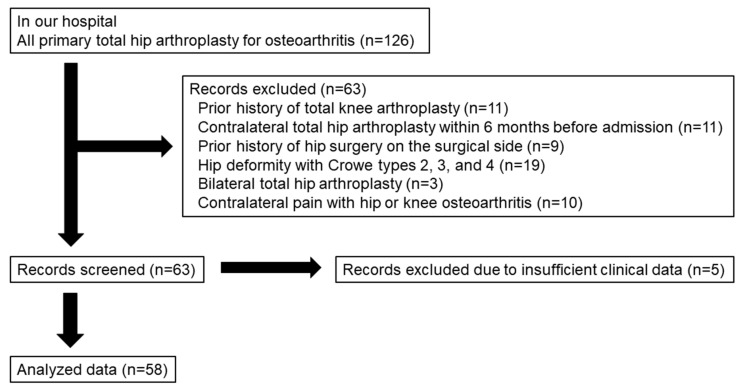
Flowchart of patient selection.

**Figure 2 jpm-13-01279-f002:**
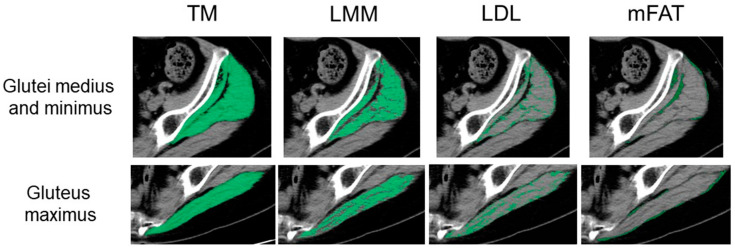
Measurement of muscle composition on an axial image of computed tomography. Total muscle cross-sectional area (TM) of the glutei medius and minimus and the upper portion of the gluteus maximus is segmented using the threshold of −29 to 150 Hounsfield units (HU). Lean muscle mass (LMM), low-density lean tissue (LDL), and intramuscular fat (mFAT) are colored as the pixels with the density of 30 to 80 HU, 0 to 29 HU, and −190 to −30 HU, respectively, within each segmented TM.

**Figure 3 jpm-13-01279-f003:**
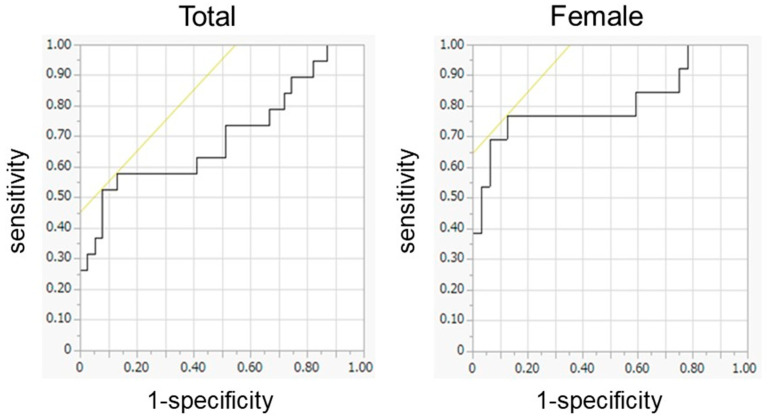
Receiver operating characteristic (ROC) curve analysis of the total cross-sectional area of the gluteus maximus in the female group (*n* = 45) and the total group of both sexes (*n* = 58) for minimum clinically important improvement in gait speed after total hip arthroplasty shown by ROC curve. The area under the curve values are the female group, 0.813, and the total group, 0.700.

**Table 1 jpm-13-01279-t001:** Demographics and clinical data before and after the operation.

	Total (*n* = 58)	Females (*n* = 45)	Males (*n* = 13)	*p* Value	Effect Size (r)
Age (years)	70.9 (9.5)	70.4 (9.0)	72.5 (11.4)	0.481 *	0.09
Body mass index (kg/m^2^)	23.0 (3.4)	22.4 (3.4)	24.9 (2.7)	**0.018 ***	0.31
Gait speed (m/s)					
Preoperative	0.87 (0.32)	0.88 (0.29)	0.81 (0.44)	0.598 *	0.07
Postoperative	1.09 (0.23)	1.07 (0.21)	1.16 (0.30)	0.295 *	0.14
Improvement	0.22 (0.28)	0.18 (0.28)	0.35 (0.25)	0.055 *	0.25
Ipsilateral knee extensor strength (Nm/kg)					
Preoperative	0.84 (0.33)	0.82 (0.32)	0.92 (0.37)	0.334 *	0.13
Postoperative	1.08 (0.41)	1.01 (0.35)	1.32 (0.48)	**0.026 ****	0.32
Contralateral knee extensor strength (Nm/kg)					
Preoperative	1.08 (0.44)	0.99 (0.36)	1.41 (0.52)	**0.006 ****	0.35
Postoperative	1.23 (0.46)	1.14 (0.41)	1.53 (0.50)	**0.016 ****	0.36
Ipsilateral hip abductor strength (Nm/kg)					
Preoperative	0.44 (0.21)	0.43 (0.21)	0.47 (0.21)	0.565 *	0.08
Postoperative	0.73 (0.27)	0.71 (0.27)	0.80 (0.30)	0.307 *	0.14
Contralateral hip abductor strength (Nm/kg)					
Preoperative	0.54 (0.24)	0.53 (0.22)	0.59 (0.29)	0.452 *	0.10
Postoperative	0.80 (0.34)	0.76 (0.31)	0.94 (0.40)	0.092 *	0.22

Data are expressed as mean (standard deviation). Gait speed improvement was calculated as the subtraction of preoperative gait speed from postoperative gait speed. *p* values at < 0.05 are shown in bold. *p* values were determined by *t* test * and by Mann–Whitney U test ** between the female and male data.

**Table 2 jpm-13-01279-t002:** Preoperative muscle composition.

	Glutei medius and minimus					Gluteus maximus				
	Total (*n* = 58)	Females (*n* = 45)	Males (*n* = 13)	*p* Value	Effect Size (r)	Total (*n* = 58)	Females (*n* = 45)	Males (*n* = 13)	*p* Value	Effect Size (r)
TM (cm^2^/m^2^)	13.4 (2.3)	12.9 (2.3)	15.2 (1.5)	**0.001 ***	0.41	9.6 (2.2)	9.3 (2.2)	10.7 (2.0)	**0.046 ***	0.26
LMM (cm^2^/m^2^)	7.6 (2.6)	7.3 (2.6)	8.8 (2.5)	0.061 *	0.25	4.6 (2.5)	4.3 (2.3)	5.8 (2.7)	**0.045 ***	0.26
LMM/ TM (%)	55.8 (13.2)	55.4 (13.5)	57.3 (12.7)	0.642 *	0.06	46.5 (18.8)	44.5 (18.1)	53.5 (20.2)	0.128 *	0.20
LDL (cm^2^/m^2^)	3.5 (0.8)	3.4 (0.9)	3.9 (1.0)	0.085 *	0.23	3.5 (1.2)	3.5 (1.1)	3.5 (1.5)	0.952 *	0.01
LDL/ TM (%)	26.6 (7.1)	26.8 (6.9)	26.0 (8.0)	0.726 *	0.05	36.9 (10.8)	38.0 (10.0)	33.3 (13.1)	0.171 *	0.18
mFAT (cm^2^/m^2^)	2.2 (1.5)	2.2 (1.5)	2.1 (1.3)	0.933 **	0.01	1.2 (0.6)	1.3 (0.6)	0.9 (0.5)	0.061 **	0.25
mFAT/ TM (%)	17.7 (14.2)	18.7 (15.2)	14.3 (9.3)	0.450 **	0.10	13.0 (7.8)	14.3 (8.1)	8.7 (4.9)	**0.021 ****	0.30
IMAT (cm^2^/m^2^)	5.7 (2.1)	5.6 (2.1)	6.0 (2.1)	0.634 **	0.06	4.6 (1.6)	4.7 (1.5)	4.4 (1.8)	0.517 *	0.09
IMAT/ TM (%)	44.3 (20.1)	45.4 (21.0)	40.3 (16.6)	0.337 **	0.13	49.9 (16.4)	52.3 (15.8)	42.0 (16.5)	**0.046 ***	0.26

Data are expressed as mean (standard deviation). *p* values were calculated by *t* test * and by Mann–Whitney test ** between the female and male data. *p* values at < 0.05 are shown in bold. TM, segmented total muscle cross-sectional area; LMM, lean muscle mass area; LDL, low-density lean tissue area; mFAT, intramuscular fat area; and IMAT, intramuscular adipose tissue area.

**Table 3 jpm-13-01279-t003:** Pearson’s correlation coefficients of gait speed with preoperative muscle composition and strength.

Muscle Composition						
	Total			Female		
	Admission	6 Months	Improvement	Admission	6 Months	Improvement
Glutei medius and minimus						
TM	0.010	0.007	−0.005	0.032	−0.118	−0.121
LMM	**0.283 ***	0.158	−0.199	**0.363 ***	0.101	**−0.301 ***
LMM/TM	**0.410 ****	0.238	**−0.280 ***	**0.505 ****	0.245	**−0.341 ***
LDL	**−0.440 ****	**−0.304 ***	**0.261 ***	**−0.532 ****	**−0.431 ****	0.232
LDL/TM	**−0.428 ****	**−0.301 ***	0.249	**−0.529 ****	**−0.333 ***	**0.301 ***
mFAT	**−0.326 ***	−0.224	0.195	**−0.388 ****	−0.263	0.207
mFAT/TM	−0.250	−0.210	0.116	**−0.302 ***	−0.220	0.149
IMAT	**−0.420 ****	**−0.289 ***	0.250	**−0.501 ****	**−0.368 ***	0.246
IMAT/TM	**−0.326 ***	−0.254	0.170	**−0.391 ****	−0.268	0.206
Gluteus maximus						
TM	**0.486 ****	**0.418 ****	−0.219	**0.569 ****	**0.346 ***	**−0.333 ***
LMM	**0.514 ****	**0.471 ****	−0.207	**0.535 ****	**0.416 ****	−0.245
LMM/TM	**0.392 ****	**0.390 ****	−0.132	**0.392 ****	**0.372 ***	−0.130
LDL	0.012	0.006	−0.009	0.158	0.043	−0.132
LDL/TM	**−0.326 ***	**−0.294 ***	0.136	**−0.307 ***	−0.246	0.135
mFAT	−0.147	**−0.269 ***	−0.052	−0.108	−0.272	−0.091
mFAT/TM	**−0.358 ***	**−0.407 ****	0.079	**−0.401 ****	**−0.409 ****	0.112
IMAT	−0.05	−0.103	−0.028	0.071	−0.084	−0.137
IMAT/TM	**−0.385 ****	**−0.387 ****	0.127	**−0.399 ****	**−0.365 ***	0.143
Muscle strength						
	Total			Female		
	Admission	6 months	Improvement	Admission	6 months	Improvement
Knee extensor						
Ipsilateral	**0.387 ****	**0.383 ****	−0.133	**0.407 ****	**0.354 ***	−0.158
Contralateral	**0.330 ***	**0.437 ****	−0.021	**0.318 ***	0.272	−0.128
Hip abductor						
Ipsilateral	**0.320 ***	**0.293 ***	−0.130	0.211	0.150	−0.107
Contralateral	**0.303 ***	**0.319 ***	−0.087	0.210	0.171	−0.091

Pearson’s correlation coefficients at *p* < 0.05 are shown in bold. TM, segmented total muscle cross-sectional area; LMM, lean muscle mass area; LDL, low-density lean tissue area; mFAT, intramuscular fat area; and IMAT, intramuscular adipose tissue area. * *p* < 0.05, ** *p* < 0.01.

**Table 4 jpm-13-01279-t004:** Stepwise regression analysis for postoperative improvement in gait speed.

	Independent Variable	B	SE (B)	β	t	*p*	95% CI	Adjusted R^2^
Total	Glutei medius and minimus LMM/TM	−0.006	0.003	−0.280	−2.185	0.033	−0.011, 0.000	0.062
Female	Glutei medius and minimus LMM/TM	−0.007	0.003	−0.341	−2.381	0.022	−0.013, −0.001	0.117

B, partial regression coefficient; SE, standard error; b, standardized partial regression coefficient; t, t-ratio; CI, confidence interval; *p*, *p* value; R^2^, coefficient of determination; TM, segmented total muscle cross-sectional area; and LMM, lean muscle mass area.

**Table 5 jpm-13-01279-t005:** Receiver operating characteristic curve analysis for minimum clinically important improvement of gait speed at ≥0.32 m/s using preoperative muscle composition and strength.

Muscle Composition						
	Total			Female		
	AUC	*p* Value	SE (95% CI)	AUC	*p* Value	SE (95% CI)
Glutei medius and minimus						
TM	0.502	0.980	0.080 (0.344, 0.660)	0.565	0.499	0.092 (0.385, 0.745)
LMM	0.599	0.223	0.079 (0.444, 0.754)	0.707	**0.031**	0.081 (0.547, 0.866)
LMM/TM	0.652	0.062	0.075 (0.505, 0.799)	0.740	**0.012**	0.078 (0.587, 0.894)
LDL	0.648	0.070	0.074 (0.504, 0.792)	0.675	0.068	0.084 (0.512, 0.839)
LDL/TM	0.655	0.058	0.074 (0.509, 0.800)	0.728	**0.017**	0.079 (0.574, 0.882)
mFAT	0.629	0.114	0.079 (0.475, 0.783)	0.700	**0.038**	0.083 (0.537, 0.862)
mFAT/TM	0.611	0.172	0.078 (0.458, 0.765)	0.700	**0.038**	0.081 (0.541, 0.858)
IMAT	0.653	0.060	0.075 (0.507, 0.799)	0.709	**0.029**	0.082 (0.549, 0.869)
IMAT/TM	0.640	0.086	0.075 (0.492, 0.787)	0.736	**0.014**	0.077 (0.584, 0.887)
Gluteus maximus						
TM	0.700	**0.014**	0.081 (0.542, 0.858)	0.813	**0.001**	0.085 (0.647, 0.978)
LMM	0.679	**0.028**	0.076 (0.530, 0.827)	0.748	**0.010**	0.073 (0.605, 0.890)
LMM/TM	0.586	0.293	0.083 (0.423, 0.748)	0.615	0.229	0.089 (0.440, 0.790)
LDL	0.584	0.300	0.093 (0.401, 0.767)	0.623	0.202	0.103 (0.421, 0.824)
LDL/TM	0.592	0.256	0.084 (0.429, 0.756)	0.615	0.229	0.091 (0.437, 0.794)
mFAT	0.528	0.734	0.089 (0.353, 0.703)	0.541	0.670	0.106 (0.332, 0.749)
mFAT/TM	0.568	0.403	0.088 (0.395, 0.741)	0.601	0.293	0.104 (0.397, 0.805)
IMAT	0.587	0.285	0.098 (0.396, 0.779)	0.625	0.193	0.111 (0.408, 0.842)
IMAT/TM	0.591	0.263	0.084 (0.427, 0.755)	0.618	0.220	0.095 (0.431, 0.804)
Muscle strength						
	Total			Female		
	AUC	*p* value	SE (95% CI)	AUC	*p* value	SE (95% CI)
Knee extensor						
Ipsilateral	0.586	0.293	0.082 (0.425, 0.746)	0.603	0.282	0.092 (0.423, 0.784)
Contralateral	0.511	0.888	0.081 (0.353, 0.670)	0.529	0.764	0.090 (0.353, 0.704)
Hip abductor						
Ipsilateral	0.530	0.709	0.085 (0.363, 0.698)	0.555	0.565	0.098 (0.364, 0.747)
Contralateral	0.552	0.524	0.084 (0.387, 0.717)	0.555	0.565	0.099 (0.361, 0.750)

*p* values at < 0.05 are shown in bold. AUC, area under the curve; SE, standard error; CI, confidence interval; TM, segmented total muscle cross-sectional area; LMM, lean muscle mass area; LDL, low-density lean tissue area; mFAT, intramuscular fat area; and IMAT, intramuscular adipose tissue area.

## Data Availability

The data used to support the findings of the present study are available from the corresponding author upon request.

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
