# Peer review of "Preoperative Lower-Limb Muscle Predictors for Gait Speed Improvement after Total Hip Arthroplasty for Patients with Osteoarthritis"

_jpm, 2023, doi:10.3390/jpm13081279_

Round 1
Reviewer 1 Report
Thank you for the oppurtinity to review this study. I enjoyed reading it. Its actually addresses a common feature of arthroplasty surgery which is muscle atrophy and rehab. But it is an original way to investigate it.
I have a few comments:
1. I would prefer using MRI to investigate muscle fibers quality and fatty infiltration. Im not sure how accurate the results are with a CT scan. It would also enable you to perform multiple f/u without the concern of radiation. with the big downside of cost, time consumption and probably availability of MRI
2. I agree that examiminig the CL side woud add some benefit to the study
3. I think that being mono centric and standart approach making the study stronger
Reviewer 2 Report
Well designed and illustration study.
Need to add a flowchart;
Discussion about the anatomical association with LMM/TM of the glutei medius and minimus and clinical outcome should be added.
